# Identifying Inequity in Treatment Allocation

Yewon Byun [1]   Dylan Sam [1]   Angel N. Desai [2]   Zachary Chase Lipton [1]   Bryan Wilder [1]

## Abstract

Disparities in resource allocation, efficacy of care, and patient outcomes along demographic lines have been documented throughout the healthcare system. In order to reduce such health disparities, it is crucial to quantify uncertainty and biases in the medical decision-making process. In this work, we propose a novel setup to audit inequity in treatment allocation. We develop multiple bounds on the treatment allocation rate, under different strengths of assumptions, which leverage risk estimates via standard classification models. We demonstrate the effectiveness of our approach in assessing racial and ethnic inequity of COVID-19 outpatient Paxlovid allocation. We provably show that for all groups, patients who would die without treatment receive Paxlovid at most 53% of the time, highlighting substantial under-allocation of resources. Furthermore, we illuminate discrepancies between racial subgroups, showing that Black patients who would die without treatment receive Paxlovid at most 32% and 65% lower than White and Asian patients, respectively.

## 1. Introduction

Long-standing evidence attests to disparities in the healthcare system, along demographic lines (e.g., race and ethnicity) (Nelson, 2002; Artiga et al., 2020; Buchmueller & Levy, 2020). For instance, recent studies have reported racial and ethnic disparities for COVID-19 outpatient treatment, for both oral antiviral drug treatment and monoclonal antibody (mAb) treatment (Boehmer, 2022; Tarabichi et al., 2023). In particular, treatment rates for Paxlovid (Nirmatralvir-Ritonavir), an oral anti-viral drug important to prevent mortality in high-risk individuals, are reported to be much lower amongst Black and Hispanic patients than White and non-Hispanic patients (Boehmer, 2022).

To reduce such health disparities, it is essential to identify biases in medical decision-making processes. However, this is fundamentally a causal inference problem that requires untestable assumptions, as we cannot simultaneously observe the impacts of treatment decisions and their counterfactual outcomes. Furthermore, it is hard to identify how the predictions of decision-makers are systematically biased since the information of decision-makers is unknown to researchers. For instance, clinicians have information during treatment assignments, not observed by researchers in electronic health records, and naive comparison of ML predictions to clinician decisions neglects this key source of uncertainty.

We address uncertainty quantification for medical decision making by developing provably valid strategies for using machine learning models to quantify inequities or inefficiencies in medical resource allocation. The output is a set of bounds on the rate at which treatment is allocated to high-risk patients, allowing users to pinpoint interpretable subgroups where treatment inequity is present after controlling for clinically meaningful covariates.

In general, producing valid bounds is difficult due to the presence of unobserved confounders, but in this setting, we leverage the unique set of circumstances for a newly introduced therapeutic (i.e., Paxlovid). We introduce a new design, based on pre-treatment (i.e., data before Paxlovid release) and post-treatment availability data (i.e., data after Paxlovid release) that allows principled auditing of allocative equity, where the allocation should only depend on one potential outcome (e.g., mortality risk without treatment). In this setting, we obtain bounds that can be estimated via standard classification models for outcomes and treatment.

We demonstrate the efficacy of this setup, by identifying racial and ethnic disparities in Paxlovid allocation for high risk individuals in the COVID-19 outpatient setting. Furthermore, this framework can generally be applied to settings with pre-availability and post-availability data, such as the introduction of other new therapeutics, the creation of new services or government programs, etc.

---

[1]Department of Machine Learning, Carnegie Mellon University [2]Department of Internal Medicine, University of California, Davis. Correspondence to: Yewon Byun <yewonb@cs.cmu.edu>.

*Workshop on Interpretable ML in Healthcare at International Conference on Machine Learning (ICML)*, Honolulu, Hawaii, USA. 2023. Copyright 2023 by the author(s).

## 2. Relevant Work

**Unobserved Confounding and Uncertainty Quantification.** Many approaches in statistics and machine learning tackle the problem of handling unobserved and potentially confounding covariates. One approach is through a marginal structural model (Robins et al., 2000; Hernán et al., 2001; Brumback et al., 2004; Joffe et al., 2004; Bonvini et al., 2022), which captures the impacts of interventions on resulting outcomes, by estimating time-dependent weights on the confounding variables. Another similar approach is through propensity scores (Tan, 2006; Kallus & Zhou, 2021), which models the likelihood of receiving a treatment given a particular set of observed features. We use similar notions from this line of work (and particularly from Tan (2006)), to regulate the impact of unobserved covariates, which helps in our search for tighter bounds: a key quantity in our showing of inequity in treatment allocation. While less related, Kallus & Zhou (2019) proposes minimax optimal policies with unobserved confounders, providing bounds on their impact through observable quantities. Finally, some work focuses on the impacts of unobserved confounders on *individual* treatment effects and proposes their own conformal prediction intervals (Lei & Candès, 2021; Jin et al., 2023).

**Studying Treatments and Decision-Making.**

There has been considerable work that studies decision-making and its relation to policies learned by machine learning models (Kleinberg et al., 2018; Mullainathan & Obermeyer, 2022). Rambachan (2021) provides a broad statistical testing framework that can detect systematic mistakes in the presence of unobserved variables, empirically identifying bias in pretrial release decisions of New York judges. Machine learning models that produce risk estimates have been used as aides in decision-making in healthcare (Caruana et al., 2015) as well as other scenarios (Kehl & Kessler, 2017; Chouldechova et al., 2018; Wilder et al., 2021). In the setting of no unobserved confounding variables, Coston et al. (2020) proposes new metrics that are potentially more helpful in decision-making. Our work uses machine learning models to produce estimates of mortality risk, which we use to analyze treatment allocation to different subpopulations with the same risk estimates.

## 3. Preliminaries

We observe a set of pre-treatment labeled data $\mathcal{X} = \{(x_i, \tau_i, g_i, y_i)\}_{i=1}^n$ and a set of post-treatment labeled data $\mathcal{X}' = \{(x_i', \tau_i', g_i', y_i')\}$, where $\tau$ is the treatment variable, $g$ denotes the group attribute for that individual, and $y$ denotes mortality. Our goal is to assess inequity in treatment allocation, utilizing pre-treatment and post-treatment availability data (see Section 4). To do so, we use standard classification models (i.e., XGBoost, Logistic Regression) to produce risk

estimates of mortality. As such, we consider a binary classification setting where we want to learn a classifier $f : \mathcal{X} \to \{0, 1\}$ to predict mortality $y$ and a classifier $h : \mathcal{X}' \to \{0, 1\}$ to predict treatment $\tau$. We also use the standard notion of a potential mortality outcome where $y(1)$ or $y(0)$ denotes the outcome of a patient that did or did not receive treatment.

We make the covariate shift assumption, where the pre-treatment distribution ($p_{\text{pre}}$) and post-treatment distribution ($p_{\text{post}}$) can have different marginal distributions over the covariates, although the labeling functions must stay the same. More formally, this has that

$$p_{\text{pre}}(y|x) = p_{\text{post}}(y|x), p_{\text{pre}}(x) \neq p_{\text{post}}(x).$$

This assumption ensures that a model that is trained on each distribution will still classify examples as the same on either distribution. For example, we train a model $h$ on $p_{\text{post}}$ (i.e., data after treatment release) that predicts treatment and apply this model on $p_{\text{pre}}$ (i.e., data before treatment release). We have that $p_{\text{pre}}(\tau = 1|x) = 0$. Thus, a model trained on this data would always allocate no treatment. However, we want our model to give the same prediction on $p_{\text{pre}}$ as it would on $p_{\text{post}}$. Intuitively, we want our model to produce the same treatment decision that a doctor would have made if the drug was available in the pre-treatment period.

## 4. Analysis of Treatment Allocation Inequity

Our goal is to assess whether or not our treatment of interest is allocated equally for patients of similar risk over different subpopulations. In order to evaluate this, we propose the following as a measure of allocation equity:

$$|P(\tau = 1|y(0) = 1, g = \alpha) - P(\tau = 1|y(0) = 1, g = \beta)|, \quad (1)$$

for different subgroups $\alpha$ and $\beta$. If this difference is large, we can conclude that treatment $\tau$ is being allocated unequally to different groups $g$, given the same level of mortality $y(0)$. We note that our proposed metric assumes all subgroups respond similarly to our treatment of interest.

We note that it is not possible to directly estimate this quantity since it conditions on $y(0)$. We can never simultaneously observe $\tau = 1$ and $y(0)$, making this a causal inference problem. Computing this directly would require an ignorability condition, or that $y \perp \tau|x$. In other words, ignorability assumes independence of treatment assignment and potential outcomes, which controls for all confounding variables. We remark that this is a much stronger assumption than what is required by our pre/post-treatment setup; these two periods are similar in that nothing has changed in the medical context, while ignorability is implausible in most real-world contexts.

We develop bounds under three different strengths of assumptions, which each yield informative results.

### 4.1. No Additional Assumptions

Without any additional assumptions on confounders or the treatment assignment process, we observe the following bound on these quantities of interest. We remark that, in this setting of arbitrary unobserved confounding, we surprisingly obtain non-vacuous, informative bounds (see Section 5.2.1).

**Proposition 1.**

$$P(\tau = 1|y(0) = 1, x) \leq \frac{P(\tau = 1|x)}{P(y(0) = 1|x)}$$

$$P(\tau = 1|y(0) = 1, x) \geq \frac{P(\tau = 1|x) - P(y(0) = 0|x)}{P(y(0) = 1|x)}$$

*Proof.* This holds as

$$P(\tau = 1|x) = P(y(0) = 1|x)P(\tau = 1|y(0) = 1, x)$$
$$+ P(y(0) = 0|x)P(\tau = 1|y(0) = 0, x)$$

We can rearrange this equation, giving us that

$$P(\tau = 1|y(0) = 1, x) =$$
$$\frac{P(\tau = 1|x) - P(y(0) = 0|x)P(\tau = 1|y(0) = 0, x)}{P(y(0) = 1|x)}$$

Then, we observe that $0 \leq P(\tau = 1|y(0) = 0, x) \leq 1$, which gives us that

$$P(\tau = 1|y(0) = 1, x) \leq \frac{P(\tau = 1|x)}{P(y(0) = 1|x)},$$

$$P(\tau = 1|y(0) = 1, x) \geq \frac{P(\tau = 1|x) - P(y(0) = 0|x)}{P(y(0) = 1|x)}.$$

$\square$

We can estimate the quantities $P(\tau = 1|x)$ and $P(y(0) = 1|x)$ by *training classifiers* on post-treatment and pre-treatment data respectively, and then evaluate them on a held-out set of data from the pre-treatment period. We note that we train classifier $f$ for mortality prediction on pre-treatment data, because we do not want our treatment of interest $\tau$ to influence $y$.

We note that

$$P(\tau = 1|y(0) = 1) = \sum_x P(\tau = 1|y(0) = 1, x) \cdot P(x|y(0) = 1)$$

Therefore, we average the bounds computed in Proposition 1 over only those in the held-out test set with $y(0) = 1$.

### 4.2. Bounded Confounding Assumption

In order to construct a tighter bound on the values in Equation 1, we must address the influence of unobserved variables on the task of interest. We introduce a parameter $\gamma$ that captures the extent of the impact of these unobserved confounders on $y(0)$, similar to the setup from prior work (Tan,

2006). This model allows us to, under an assumption that confounding is limited, assess whether there are verifiable discrepancies in treatment rates across subgroups. With this framework, we can vary $\gamma$ over a range of values to determine to how much confounding our finding is robust. More formally, we assume that $\exists \gamma$ s.t.

$$\frac{1}{\gamma} \leq \frac{P(y(0) = 1|\tau = 0, x)}{P(y(0) = 1|\tau = 1, x)} \leq \gamma. \tag{2}$$

We note that $\gamma = \infty$ is equivalent to making no assumptions, or arbitrary confounding. In this scenario, we can recover the result in Proposition 1. Thus, with this term $\gamma$, we observe the following inequality:

**Proposition 2.**

$$\frac{1}{\gamma\left(\frac{P(\tau=0|x)}{P(\tau=1|x)}\right) + 1} \leq P(\tau = 1|y(0) = 1, x) \leq \frac{1}{\frac{1}{\gamma}\left(\frac{P(\tau=0|x)}{P(\tau=1|x)}\right) + 1}$$

*Proof.* Deferred to Appendix A. $\square$

We can again compute the quantities $P(\tau = 0|x)$ and $P(\tau = 1|x)$ on the same held-out set of pre-treatment data from Section 4.1, with a treatment predictor trained on post-treatment data. We again average this quantity over patients in the hold-out set with $y(0) = 1$.

We compute this bound per racial and ethnic group, with varying parameters of $\gamma \in [1, 2]$, which represents a range from no unobserved confounding ($\gamma = 1$) to a moderate degree of confounding ($\gamma = 2$). We provide a frame of reference for values of $\gamma$ in Section 5.2.3.

### 4.3. Bayes Optimality Assumption

In this setting, we test an assumption that doctors assign treatments according to a Bayes optimal strategy (that at least accesses all variables observed by us). While doctors are domain experts, they may not always follow a Bayes optimal policy. If doctors are indeed Bayes optimal, we observe the following inequality:

$$P(\hat{\tau} = 1|y(0) = 1, x) \leq P(\tau = 1|y(0) = 1, x),$$

where $\hat{\tau}$ is some policy that only uses observed covariates $x$.

**Proposition 3.** *Let $\hat{\tau}$ be some treatment policy, and let $u$ be a set of unobserved covariates. Then,*

$$P(\hat{\tau} = 1|y(0) = 1, x) \leq P(\tau = 1|y(0) = 1, x, u).$$

The underlying intuition is that, with only the observed covariates, we can do no better than a Bayes optimal policy that accesses unobserved covariates. In other words, the doctor's strategy is more effective than our approximation

that uses only the observable variables. This is rather intuitive as an optimal strategy should more frequently assign treatment to patients with $y(0) = 1$ or that would die without any treatment.

We consider the following treatment policy that is a lower bound to a Bayes optimal policy over the observed covariates. Let $\hat{y} \in \{0, 1\}$ denote the output of our classifier $f$ trained on $\mathcal{X} \sim D_{\text{pre}}$. Let $\hat{r}(x)$ denote our classifier's predicted probabilities for $y$, where $\hat{r}(x) \in [0, 1]$. We assign $\tau = 1$ to samples with the top $P_{\mathcal{X} \sim D_{\text{post}}}(\tau = 1)$ fraction of risk values $\hat{r}(x)$, and $\tau = 0$ otherwise. We evaluate this policy on a held-out set of pre-treatment test data (see Section 5.2.4).

# 5. Results for Inequity in Paxlovid Allocation

We apply our analysis framework to understand treatment allocation inequity and efficiency of Paxlovid (nirmatrelvir-ritonavir) in the COVID-19 outpatient setting.

## 5.1. Dataset and Cohort Definition

We use the NCATS NC3 cohort (Haendel et al., 2020), consisting of national line-level data of $18,438,581$ total patients, including $7,149,421$ confirmed COVID-19 positive patients and $198,717$ possible COVID-19 positive patients, pooled from 76 different data sharing centers across the United States. We focus our analysis on outpatients with a positive SARS-CoV-2 test result, satisfying eligibility requirements outlined in Appendix C. Due to underreported figures for prescriptions at certain sites, we restrict our cohort to sites with at least a $10\%$ treatment rate.

## 5.2. Results

### 5.2.1. ARBITRARY UNOBSERVED CONFOUNDING

We observe the following bounds for different racial and ethnic groups. We remark that even without placing any additional assumptions on confounding ($\gamma = \infty$), we obtain non-vacuous, informative bounds:

$$
\begin{aligned}
0.039 \leq \quad & P(\tau = 1 | y(0) = 1, x, g_{hisp}) \quad && \leq 0.429 \\
0.089 \leq \quad & P(\tau = 1 | y(0) = 1, x, g_{not\_hisp}) \quad && \leq 0.375 \\
0.094 \leq \quad & P(\tau = 1 | y(0) = 1, x, g_{white}) \quad && \leq 0.387 \\
0.066 \leq \quad & P(\tau = 1 | y(0) = 1, x, g_{black}) \quad && \leq 0.348 \\
0.102 \leq \quad & P(\tau = 1 | y(0) = 1, x, g_{asian}) \quad && \leq 0.535
\end{aligned}
$$

These bounds show that for all groups, there is a significant under-allocation of resources since patients who would *die* without treatment receive Paxlovid at most $53\%$ of the time. To assess inequity between subgroups, we need to construct tighter and non-overlapping bounds across subgroups. Thus, we move our setting to assume some $\gamma$-measure of unobserved confounding.

### 5.2.2. $\gamma$-MEASURE OF UNOBSERVED CONFOUNDING

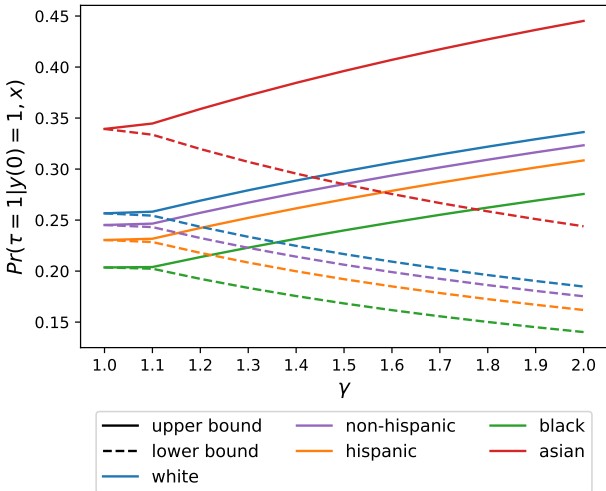

*Figure 1.* Upper and lower bounds for quantity of interest $P(y = 1 | \tau = 1, x)$ is computed for each racial/ethnic group, with varying values of $\gamma \in [1, 2]$. Post-treatment availability data (post December 22, 2021) is used to compute quantities. Solid lines represent upper bounds and dashed lines represent lower bounds.

Assuming $\gamma$-measure of unobserved confounding in treatment, we are able to identify non-overlapping bounds for our quantity of interest $P(\tau = 1 | y(0) = 1, x)$. We identify non-overlapping bounds between 1) Blacks and Asians up to high values of $\gamma \approx 1.8$; 2) Blacks and Whites ($1 \leq \gamma \leq 1.37$); and 3) Hispanics and non-Hispanics ($1 \leq \gamma \leq 1.15$). We clearly identify that treatment allocation rates for Black patients that would *die* without treatment are at most $32.00\%$ ($\gamma = 1.37$) and $64.88\%$ lower ($\gamma = 1.8$) than treatment rates for White and Asian patients, respectively. Similarly, we can observe that treatment rates for Hispanic patients that would die without treatment are at most $6.45\%$ ($\gamma = 1.15$) lower than treatment rates for non-Hispanic patients. Further, we uncover a high rate of underdiagnosis across all populations, for all values $\gamma \in [1, 2]$.

### 5.2.3. INTERPRETATION OF $\gamma$

Our results identify provable inequity under specific values of $\gamma$. Since $\gamma$ is impossible to compute in practice, the feasibility assumption is difficult to ascertain. To give a frame of reference for the strength of our assumption, we can compute an analogous $\gamma'$ for an *observed* covariate (e.g., diabetes), which *is* computable in practice. We can similarly determine this value of $\gamma'$ by training a discriminative model to compute the ratio $\frac{P(y(0)=1|z=0,x)}{P(y(0)=1|z=1,x)}$, where $z$ is the random variable that represents if the patient $x$ has the covariate of interest. We select diabetes as our covariate of interest, based on its well-documented association with high risk of severe COVID-19 (Centers for Disease Control and

Prevention, 2023). We again proceed by training a classifier $f$ to predict mortality and evaluate on a held-out test set of patients with counterfactual features of having diabetes ($z = 1$) and not having diabetes ($z = 0$). From the value of this ratio, we can then compute a similar value of

$$\frac{1}{\gamma'} \le \frac{P(y(0) = 1 | z = 0, x)}{P(y(0) = 1 | z = 1, x)} \le \gamma'$$

Taking the smallest value of $\gamma'$ that satisfies our assumptions for the influence of diabetes, we observe such a value of $\gamma'$ is 1.42. Therefore, our result in identifying disparities in allocation (for example, Blacks and Asians have non-overlapping bounds at a value of $\gamma \approx 1.8$) is robust to confounders that exhibit a **stronger influence** on COVID-19 mortality risk than the presence of a patient's diabetes, a condition evidenced to be associated with high risk of severe COVID-19 (Centers for Disease Control and Prevention, 2023).

### 5.2.4. BAYES' OPTIMALITY ASSUMPTION

To test our Bayes' optimality assumption, we empirically compute an upper bound via Proposition 1 under no additional assumptions on confounding and reconcile it with the proposed approximation of the Bayes optimal strategy. As a result, we observe that

$$P(\hat{\tau} = 1 | y(0) = 1, x) \nleq \frac{P(\tau = 1 | x)}{P(y(0) = 1 | x)}$$

or that the lower bound produced by our Bayes optimal strategy is greater than the valid upper bound. For example, we observe that our lower bound on the Bayes optimal strategy achieves an allocation rate of 78.9% for Asians, which is a much larger allocation rate than the upper bound in Proposition 1, which has an allocation rate of 53.5%. We report the full results for each group in Appendix B. As a consequence, this refutes the assumption that doctors are Bayes optimal with respect to at least as much information as we observe.

As such, this finding does not provide a viable lower bound, but rather, it makes claims about the doctors' decisions regarding the allocation of $\tau$.

## 6. Discussion

Our framework introduces a principled approach, using machine learning to audit inequity in treatment allocation. We demonstrate our approach, which makes much weaker assumptions than standard yet unrealistic ignorability conditions, to *provably* identify inequity for different subgroups in the allocation of a new treatment. We remark that this setting is quite broad; it can easily be applied to different applications such as the creation of new services, government programs, etc. Equivalently, it can be applied to policies, benefits, or treatments that roll out in one location and not the other.

## Acknowledgements

We gratefully acknowledge the NSF (FAI 2040929 and IIS2211955), UPMC, Highmark Health, Abridge, Ford Research, Mozilla, the PwC Center, Amazon AI, JP Morgan Chase, the Block Center, the Center for Machine Learning and Health, and the CMU Software Engineering Institute (SEI) via Department of Defense contract FA8702-15-D-0002, for their generous support of ACMI Lab's research.

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

## A. Additional Proofs

We now present the proof of Proposition 2.

*Proof.* Using Bayes Rule, we can rewrite the equation, giving us that

$$\frac{1}{\gamma} \leq \frac{P(\tau = 0|y(0) = 1, x)}{P(\tau = 1|y(0) = 1, x)} \cdot \frac{P(\tau = 1, x)}{P(\tau = 0, x)} \leq \gamma$$

We can rearrange this equation,

$$\frac{1}{\gamma} \cdot \frac{P(\tau = 0, x)}{P(\tau = 1, x)} \leq \frac{P(\tau = 0|y(0) = 1, x)}{P(\tau = 1|y(0) = 1, x)} \leq \gamma \cdot \frac{P(\tau = 0, x)}{P(\tau = 1, x)}$$

Substituting $P(\tau = 0|y(0) = 1, x) = 1 - P(\tau = 1|y(0) = 1, x)$ gives us that

$$\frac{1}{\gamma} \cdot \frac{P(\tau = 0, x)}{P(\tau = 1, x)} \leq \frac{1 - P(\tau = 1|y(0) = 1, x)}{P(\tau = 1|y(0) = 1, x)} \leq \gamma \cdot \frac{P(\tau = 0, x)}{P(\tau = 1, x)}$$

Let $a = P(\tau = 1|y(0) = 1, x)$. We can rearrange this equation,

$$\frac{1}{\gamma} \cdot \frac{P(\tau = 0, x)}{P(\tau = 1, x)} \cdot a \leq (1 - a) \leq \gamma \cdot \frac{P(\tau = 0, x)}{P(\tau = 1, x)} \cdot a$$

We remark that $\frac{P(\tau=0,x)}{P(\tau=1,x)} = \frac{P(\tau=0|x)}{P(\tau=1|x)}$. Taking the left-hand side, we can derive the upper bound. Similarly, we can derive the lower bound from the right-hand side.

$$\frac{1}{\gamma \left( \frac{P(\tau=0|x)}{P(\tau=1|x)} \right) + 1} \leq P(\tau = 1|y(0) = 1, x) \leq \frac{1}{\frac{1}{\gamma} \left( \frac{P(\tau=0|x)}{P(\tau=1|x)} \right) + 1}.$$

$\square$

## B. Bayes Optimal Assumption Results

We observe the following results for the lower bound generated through the Bayes optimal assumption, compared to the upper bound generated through Proposition 1.

$$P(\hat{\tau} = 1|y(0) = 1, x, g_{hisp}) = 0.5357 \nleq 0.4285 = \frac{P(\tau = 1|x, g_{hisp})}{P(y(0) = 1|x, g_{hisp})}$$

$$P(\hat{\tau} = 1|y(0) = 1, x, g_{not\_hisp}) = 0.5856 \nleq 0.3751 = \frac{P(\tau = 1|x, g_{not\_hisp})}{P(y(0) = 1|x, g_{not\_hisp})}$$

$$P(\hat{\tau} = 1|y(0) = 1, x, g_{white}) = 0.6225 \nleq 0.3872 = \frac{P(\tau = 1|x, g_{white})}{P(y(0) = 1|x, g_{white})}$$

$$P(\hat{\tau} = 1|y(0) = 1, x, g_{black}) = 0.4680 \nleq 0.3479 = \frac{P(\tau = 1|x, g_{black})}{P(y(0) = 1|x, g_{black})}$$

$$P(\hat{\tau} = 1|y(0) = 1, x, g_{asian}) = 0.7894 \nleq 0.5354 = \frac{P(\tau = 1|x, g_{asian})}{P(y(0) = 1|x, g_{asian})}$$

As such, in all cases, we observe that

$$P(\hat{\tau} = 1|y(0) = 1, x) \nleq \frac{P(\tau = 1|x)}{P(y(0) = 1|x)}.$$

This implies that our assumption about doctors' treatment policies is false; they do not appear to follow Bayes optimal strategies (with respect to risk estimates from a model on observable data).

# C. Dataset and Cohort Details

## C.1. Dataset Acknowledgement Statement

The analyses described in this publication were conducted with data or tools accessed through the NCATS N3C Data Enclave https://covid.cd2h.org and N3C Attribution & Publication Policy v 1.2-2020-08-25b supported by NCATS U24 TR002306, Axle Informatics Subcontract: NCATS-P00438-B. This research was possible because of the patients whose information is included within the data and the organizations (https://ncats.nih.gov/n3c/resources/data-contribution/data-transfer-agreement-signatories) and scientists who have contributed to the on-going development of this community resource [https://doi.org/10.1093/jamia/ocaa196].

## C.2. Cohort Details

The patient cohort is filtered out based on the following eligibility requirements:

- Satisfy all FDA-approved Paxlovid eligibility requirements (U.S. Food and Drug Administration, Year of Access)

- Not taking any medications, where coadministration with Nirmatralvir-Ritonavir is contraindicated (Marzolini et al., 2022; Larkin, 2022)

- First COVID-19 diagnosis visits are between 22 December 2021 (date of FDA approval for Paxlovid) and 31 May 2023

- From sites with at least a 10% treatment rate—to exclude sites where treatment is potentially underreported.

