# OpenReview forum: "Identifying Inequity in Treatment Allocation"
_ICML.cc/2023/Workshop/IMLH — IMLH 2023 PosterShortPaper_

### Official Review · Reviewer_uFxF · 2023-06-16
**Important problem considered**

**Rating:** 7
**Confidence:** 4

**Review:**

The authors present an approach to estimate prevailing inequalities in treatment allocation. Albeit, their presentation focuses on a clinical use-case of Paxlovid drug use for Covid infections to highlight biases in medical decision-making processes, the approach generalizes to any use-case that provides pre-/post-treatment data. The importance of this line of work should not be under-stated, as developing tools to uncover such disparities with respect to health outcomes (here mortality risk) in turn can focus attention towards mitigating potentially structural biases.

The paper is well written and easy to follow. The technical steps of the approach to developing bounds is well motivated. The metric of interest to measure inequity is the absolute difference between any two groups of the probability of receiving a treatment given the same level of mortality. They observe that calculating this quantity directly requires the ignorability assumption, which is too strong in their setting. As a consequence, they develop three steps with varying strengths on the underlying assumptions and show that under their framework inequity of Paxlovid drug treatment exists between protetcted groups.

The authors developed 3 bounding methods: 1) informative bounds without any additional assumptions on confounders or the treatment assignment process; 2) tighter bounds where varying limit of unobserved confounders can be captured; 3) the third step tests the effectiveness of the approximation assuming doctors are Bayes optimal.

The mathematical approach is clearly derived and addresses the key aspects of measuring disparity in treatment allocation.

If there was one comment, it would be great to come up with some diagnostics to support the assumptions being made. I understand through controlling the level of impact of unobserved confounders this may to some degree be addressed, but I’m still curious how unequal healthcare utilization among the protected groups could challenge the approaches been taken, e.g., when faced with missing data and when this missingness is not random. Maybe this should be couched into the covariate shift assumption being made?

---

### Official Review · Reviewer_sicx · 2023-06-18
**Solid Setup with Theoretical Justifications**

**Rating:** 7
**Confidence:** 3

**Review:**

The authors present a method to study inequalities in treatment allocation between demographic groups. They frame the problem as a causal inference problem, and present three assumptions of varying strengths in order to make it tractable. They demonstrate their method on real-world COVID-19 treatment data.

The paper is generally well-written. The proposed setup makes sense, and has strong theoretical justifications. I think it would be a solid contribution to the workshop.

I have a few suggestions and questions to potentially improve the paper:
1. The authors propose (1) as the definition of "fair" treatment allocation. I question whether this is the best definition of fairness, as it doesn't seem to take potential differences in treatment effect between groups into account. For example, if a drug is more effective in one demographic group in another, it seems natural that there would be differences in its allocation.
2. I was confused by the use of pre vs. post treatment data in the analysis.
3. For completeness, the authors should show some simple descriptive statistics of the data (e.g. the base treatment rate and mortality rate of each group)
4. Have there been any previous medical studies looking into why Asian patients receive Paxlovid at such high rate relative to Black patients after adjusting for the outcome? This seems like a significant clinical result.

---

### Official Review · Reviewer_v7H5 · 2023-06-19
**The paper deals with an interesting problem, but there are concerns about the assumption and proposed measure**

**Rating:** 5
**Confidence:** 4

**Review:**

This paper proposes a novel method to assess and address inequity in treatment allocation, specifically focusing on the racial and ethnic disparities in COVID-19 outpatient Paxlovid allocation.

While the paper deals with an interesting problem, there are some major concerns about the assumption and proposed measure:

1. The key assumption may not be valid (e.g. the labeling function for pre-treatment and post-treatment patients are the same). The underlying logics of this assumption is that the treatment only affects the outcome of the patients through changing their observed covariates. However, being treated itself should be an important covariate that would affect the patients outcome. It would be more appropriate to discuss the validity of this assumption.

2. The proposed measure for allocation equity may not be proper. The disparity of treatment across different subgroups (e.g. racial groups) not only lies in the different severity distributions across subgroups, but also in the different treatment responses across subgroups. The current proposed measure only considers the former - namely gives the subgroup more treatment if there are more severe patients in this group. However, the treatment responses in different groups are not identical; in most extreme case, if one subgroup has the most severe patients but they do not respond to the treatment, it does not make sense to allocate more treatment to this subgroup. The paper would be greatly boosted if this point can be discussed.

---

### Meta-Review · Area_Chair_62R4 · 2023-06-20

**Recommendation:** Accept (Poster)
**Confidence:** 5

**Metareview:**

Two of the three reviewers expressed positive opinions about this paper, appreciating its clarity and extensive experimentation. Nevertheless, they raised significant concerns regarding the underlying assumption and the chosen measure. Therefore, I kindly request the authors to carefully consider the identified shortcomings and ensure that these issues are thoroughly addressed in the final version.

---

### Decision · Program_Chairs · 2023-06-20

Accept (Poster Short Paper)